# Narrow-Linewidth Pr:YLF Laser for High-Resolution Raman Trace Gas Spectroscopy

**Charuka Muktha Arachchige and Andreas Muller**  *

Physics Department, University of South Florida, Tampa, FL 33620, USA; charuka@usf.edu
* Correspondence: mullera@usf.edu

**Abstract:** Spontaneous Raman gas spectroscopy, which stands out as a versatile chemical identification tool, typically relies on frequency-doubled infrared laser sources to deliver the high power and narrow linewidth needed to achieve chemical detection at trace concentrations. The relatively low efficiency and high complexity of these lasers, however, can make them challenging to integrate into field-deployable instruments. Additionally, the frequency doubling prevents the utilization of circulating laser power for Raman enhancement. A diode-pumped Pr:YLF laser was investigated as an alternative narrow-band light source that could potentially realize a more portable Raman scattering system. When operated with an intracavity etalon, the laser realized a linewidth of $0.5 \, \text{cm}^{-1}$ with a green output power of 0.37 W and circulating power of 16 W when pumped with 3.1 W from a blue diode laser. Trace detection at atmospheric pressure with a high degree of spectral discrimination was demonstrated by resolving overlapping $N_2/CO$ and $CO_2/N_2O$ Raman bands in air.

**Keywords:** Raman scattering; praseodymium laser; trace detection; multipass enhancement

## 1. Introduction

An ideal tool for optical chemical identification, spontaneous Raman scattering (SRS) requires only a single laser source and analyzer to detect a large number of molecular analytes simultaneously and with isotopologue resolution [1]. Even homonuclear molecules with forbidden optical dipole absorption transitions can be detected by SRS. However, as a light-matter interaction, SRS is relatively weak, particularly for gases, thus typically requiring a high-power laser as the pump source. Because the SRS cross-section grows with the fourth power of the Stokes frequency, a light source in green or blue is highly desirable. Unfortunately, few laser gain crystals are available in that spectral region, and thus SRS pump sources mostly consist of frequency-doubled infrared neodymium or ytterbium lasers. Simplicity and efficiency are properties that are difficult to obtain with these laser sources. For example, a conventional commercial few-Watt green laser typically consumes over hundred Watts of electrical power and has a footprint of order ten liters when the power supply is accounted for. Additionally, due to the second-harmonic generation process, the circulating intracavity power of such lasers cannot be utilized for SRS enhancement. Consequently, alternative laser sources with gain directly in the blue or green could create new opportunities for advancing SRS spectroscopy. In particular, portable and possibly integrated Raman devices could be realized with such lasers, which could enable field and consumer applications heretofore inaccessible to SRS-based chemical analyzers.

Today, the vast majority of blue or green laser sources are built on semiconductor gain materials such as indium gallium nitride. While featuring exceptional efficiency, the linewidth of diode lasers cannot typically be reduced below a few $\text{cm}^{-1}$ when the optical output power is on the order of one watt due to their small size and multimodal nature [2,3]. Specialized devices such as semiconductor tapered amplifiers can circumvent this problem, but these are currently not commercially available in the blue. Alternatively, a variety of rare-earth-doped gain media have been investigated under blue diode pumping

for lasing in the visible [4]. For instance, dysprosium-doped glasses can generate yellow light [5]. Another promising medium is praseodymium-doped yttrium-lithium-fluoride (Pr:YLF), with gain in the red, orange, green, and cyan [6]. However, although Pr:YLF lasers have substantially matured recently and output powers of several watts at 523 nm have been achieved using diode pumping [7,8], it remains to be determined whether such performance can be obtained with sufficient spectral purity and whether the associated circulating power may be utilized for SRS spectroscopy.

Here we report on the investigation of a frequency-narrowed 523 nm Pr:YLF laser for application in trace gas SRS spectroscopy. We focused on the regime in which a single, passively-cooled, InGaN laser diode continuously pumped Pr:YLF with 3.1 W at a wavelength of 443 nm, sourcing 8 W of electrical power. The laser's spectral characteristics were measured with a resolution of 0.3 cm$^{-1}$ with and without an intracavity etalon. The etalon reduced the linewidth from tens of cm$^{-1}$ down to $\approx$0.5 cm$^{-1}$ albeit at a $\approx$40% reduction in output power. Nevertheless, the spectrally narrowed light could be used for SRS gas detection either in the form of circulating power ($\approx$16 W) in a 90º scattering geometry or via 0.37 W outcoupled into an externally multipassed collinear geometry. High spectral discrimination was achieved at trace concentrations as evidenced by resolving partially overlapping $N_2$/CO and $CO_2$/$N_2O$ Raman bands in air at atmospheric pressure.

## 2. Background

The chemical trace analysis of gases plays a key role in science and industry alike, with applications, for example, in quality control of foods, pharmaceutical production, or medical diagnostics. While established techniques such as gas chromatography or optical absorption can provide record-low limits of detection, few techniques are as versatile and robust as spontaneous Raman scattering. Any molecular analyte can, in principle, be detected, providing enough spectral resolution and SRS scattering cross-section. In practice, nevertheless, considerable efforts need to be devoted to the optimal utilization of laser power and SRS light collection. Given that the SRS intensity scales with input laser power, a high power source is obviously desirable. For this reason, almost all early SRS gas spectroscopy was implemented using the blue/green argon-ion laser [1]. It offered watt-level output power, or around a hundred watts of circulating laser power, though with considerable overall power dissipation. With the development of all-solid-state green lasers, enormous gains in laser efficiency were realized. However, the need for frequency doubling to achieve visible laser light meant that intracavity circulating power could not be directly utilized. To circumvent this limitation, numerous approaches to SRS enhancement have been researched and are continually being improved. Approaches based on cavities [9–19] or on waveguides [20–25] have demonstrated the ability to routinely detect at parts-per-million (ppm) concentrations. For example, in cavity-enhanced Raman scattering (CERS), high-power green laser light obtained via frequency doubling can be introduced into an external high-finesse optical cavity to create circulating power of 1 kW [26]. However, the demands on laser and cavity stabilization are substantial in CERS, making the realization of portable instrumentation a formidable task because of the high susceptibility to temperature changes or mechanical/acoustical disturbances. Thus, a natural step forward is to implement intracavity SRS directly with a visible laser gain medium such as Pr:YLF. This material has been studied primarily for generating high-power infrared and red light and for intracavity second-harmonic ultraviolet generation [8]. Nonetheless, power scaling for green (523 nm) and potentially cyan (479 nm) lasing make it an attractive choice for gas SRS. The purpose of the present work is to explore the suitability of Pr:YLF for high-resolution SRS gas spectroscopy. The spectral properties of a diode-pumped Pr:YLF laser are first detailed with and without intracavity etalon with the goal of realizing high spectral discrimination for chemical differentiation. Raman enhancement methods are then compared, and a multipass implementation is described. An example application is demonstrated in which nearly-overlapping bands of several common analytes can be distinguished even when their concentrations are widely different.

### 3. Laser Configuration and Characterization

#### 3.1. Design and Components

The Pr:YLF laser we investigated was built following previously explored designs. The single-end-pumped configuration realized in the form of a folded plano-concave standing-wave cavity is depicted in Figure 1. With low power consumption and portable applications in mind, we opted for an implementation in which the pump laser diode (Nichia NUBM44) was passively air-cooled and operated at a current of 2 A. At this current, the electrical power consumption equaled 8 W, which, if needed, could be supplied by a light battery for an extended period of time.

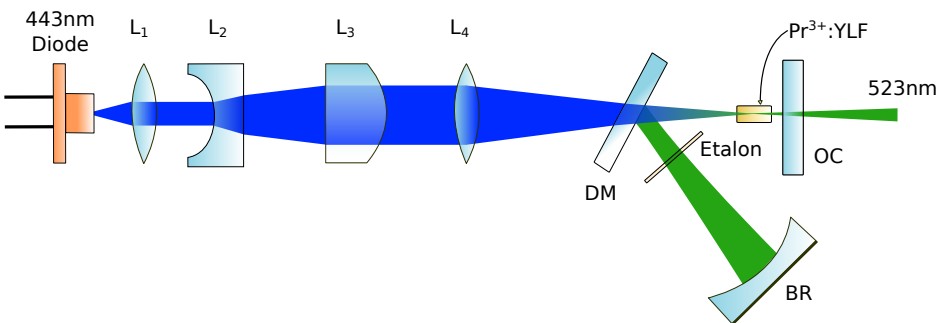

**Figure 1.** Laser configuration. The light from a multimode blue laser diode (Nichia NUBM44) is focused into a Pr:YLF crystal through a dichroic mirror (DM). A thin etalon reduces the linewidth of the emission generated near 523 nm. The threshold pump power was 0.9 W without the etalon.

The pump diode, operating near ambient temperature (typically 22 °C), generated light at a wavelength of 443 nm. It was selected for maximal absorption (>90%) by a 6 mm long, 0.5% doped, Pr:YLF gain crystal (Optogamma). Focusing of the pump beam was performed with a pair of achromatic lenses ($L_4$) which in combination had a focal length of about 45 mm. Prior to focusing, the beam emerging from the pump diode was first collimated along the fast axis by an aspheric lens ($L_1$) and subsequently enlarged along the slow axis by a cylindrical beam expander ($L_2$ and $L_3$). The pump was introduced through a dichroic fold mirror (DM) which was reflective for green but transmissive for blue light. A concave back reflector (BR) and planar output coupler (OC) with a transmission of 2.3%, separated by ≈50 mm, defined the standing-wave laser cavity.

#### 3.2. Spectral Properties

Table 1 summarizes the properties of the various components. The cavity optics had high losses in the orange and red to avoid lasing at $Pr^{3+}$ transitions other than the green. A 0.3 mm thick uncoated fused silica etalon (LightMachinery) was inserted into the intracavity beam path. Figure 2 shows the spectrum of the pump laser diode and the spectrum of the Pr:YLF laser without and with the intracavity etalon in place. The spectra were acquired with a 1 m grating spectrometer (McPherson) with a resolution of about 0.3 cm$^{-1}$. As can be seen from the pump laser diode spectrum (Figure 2a), the InGaN diode is ideally suited to pump the Pr:YLF laser, nearly matching the absorption bandwidth of several nanometers. Without the intracavity etalon (Figure 2b), the Pr:YLF laser spectrum features multiple peaks whose exact spectral location and magnitude sensitively depend on the laser's alignment and temperature. When the etalon is inserted (Figure 2c), the bandwidth narrowed to approximately 0.5 cm$^{-1}$. The insertion of the etalon also reduces the output power from 0.53 W to 0.37 W.

**Table 1.** Summary of parameters of components for Pr:YLF laser. Radii of curvature (RC), power reflectivity (*R*), and focal lengths (*f*) are indicated where relevant. "AR coated" stands for anti-reflection coated.

| Component | Description | Dim. Characteristics | Properties |
|---|---|---|---|
| $Pr^{3+}$:LiYF$_4$ crystal | Gain medium | $6 \times 3 \times 3$ mm$^3$ | 0.5% a-cut, AR coated |
| Back reflector | Concave high reflector | RC = 50 mm | R>99.98% at 523 nm |
| Output coupler | Flat partial reflector | RC = $\infty$ | R = 97.7% at 523 nm |
| Fold mirror | Dichroic reflector | RC = $\infty$ | transmit 443 nm, reflect 523 nm |
| Pump beam expander | Cylindrical 6$\times$ expander | L$_2$-L$_3$ separation 30 mm | AR coated |
| Pump lens | Focusing achromat | $f \approx 45$ mm | AR-coated pair |
| Etalon | Fused silica plate | 0.3 mm thick | uncoated, FSR = 11.2 cm$^{-1}$ |

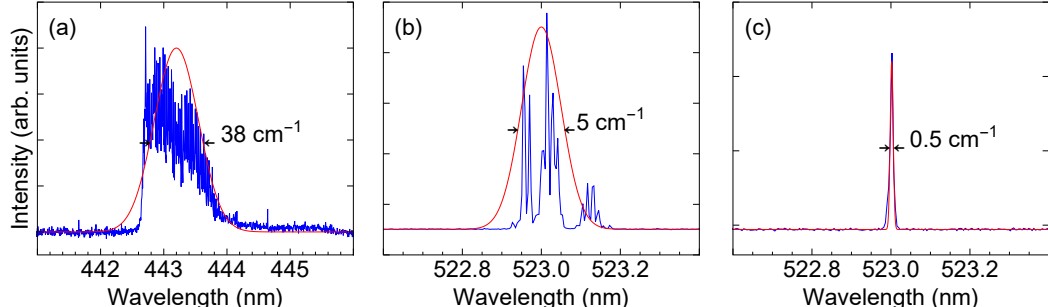

**Figure 2.** (**a**) Spectrum of pump diode. Laser output spectrum measured with a resolution of 0.3 cm$^{-1}$ without (**b**) and with (**c**) the intracavity etalon in place.

The spectrum in Figure 2 shows that, even with the etalon in place, the laser is still operating on numerous longitudinal resonator modes since the laser cavity free spectral range, FSR = $c/2L \approx 0.1$ cm$^{-1}$, is smaller than the cavity linewidth. This could be potentially remedied by the utilization of a second, thick etalon. In fact, a single longitudinal operation with a linewidth of 30 MHz has been demonstrated for the red transition in Pr:YLF under certain operating conditions [27]. However, the cost in output power would not warrant such a reduction in linewidth for applications in spontaneous Raman scattering because rotationally-resolved lines will typically be at least 0.1 cm$^{-1}$ wide anyway under ambient conditions [1].

*3.3. Spatial Characteristics*

Another important laser characteristic is the output beam's spatial profile because it can serve as a measure of spatial coherence. Importantly, the degree of spatial coherence dictates the efficiency with which the beam could be coupled into, for example, an optical resonator. We have characterized the lateral beam profile by direct imaging with a two-dimensional array detector. Figure 3 shows an image of the beam profile after heavy attenuation. As can be seen, the beam's profile is somewhat elliptical but otherwise resembles that of a Gaussian TEM$_{00}$ mode along each axis. The ellipticity could be removed if desired by, e.g., an anamorphic prism pair. To determine the degree of overlap with the latter a Gaussian TEM$_{00}$ mode, we measured the beam parameters via focusing by an achromatic lens. By measuring the beam spot size as a function of position, we extracted the beam divergence in relation to the spot size at the focus. As depicted in Figure 3b, such a measurement enables the extraction of an average M$^2$ factor of 1.1. This factor is on par with those of previously reported Pr:YLF lasers. Achieving M$^2$ = 1 would require lasing on a single longitudinal mode. However, this is not critical for spontaneous Raman enhancement, with the notable exception of high-finesse CERS. The M$^2$ factor of 1.1 and linewidth of 0.5 cm$^{-1}$ are perfectly suited for intracavity SRS enhancement as well as for multipass SRS enhancement, as we show below.

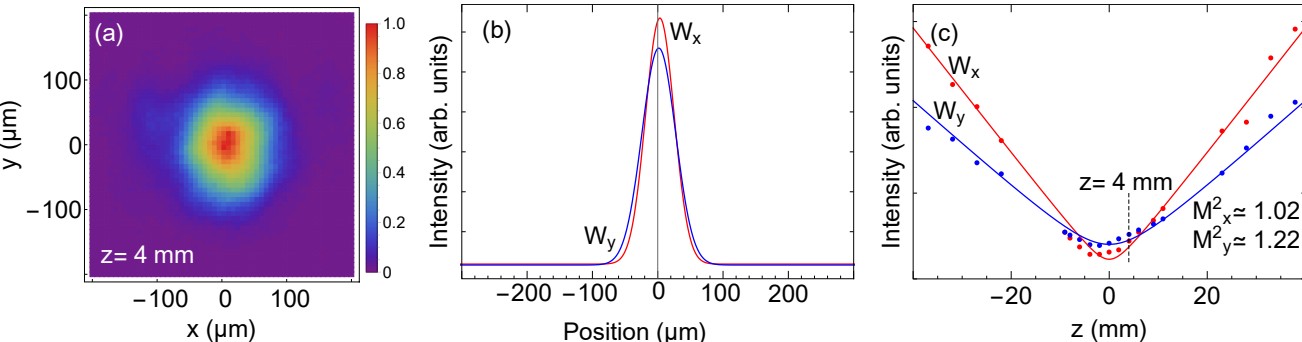

**Figure 3.** Spatial laser beam characteristics upon focusing by an achromatic lens. (**a**) Image of beam profile at position *z* = 4 mm relative to the waist location. (**b**) Cross-sectional plots along *x* and *y* directions. (**c**) Beam spot size as a function of propagation distance.

## 4. Raman Spectroscopy with Pr:YLF Laser

The general capability needed to implement spontaneous Raman enhancement is the creation of a high photon flux within a localized volume of gas. This capability is quite different from that needed in gas absorption spectroscopy, where the maximization of the propagation distance (pathlength) of a beam is required, regardless of whether the constituent beams in, e.g., a Herriott cell overlap or not.

### 4.1. SRS Enhancement Variants

The most obvious route to SRS enhancement is to utilize the high circulating power that exists within a laser cavity and collect the scattered light in a focused region of the intracavity beam. As illustrated in Figure 4a, only minor modifications to the setup in Figure 1 are needed in principle to take advantage of this enhancement. The collection optics (lens $L_6$) must be matched to the intracavity beam so as to maximize light collection. In practice, however, care must be taken to avoid background light at or near the frequency of the Raman scattered light. The near-orthogonal excitation/detection geometry depicted in Figure 4a effectively rejects such stray background light. However, a significantly higher collection of SRS photons is achieved when collecting collinearly, as shown in Figure 4b. In that case, however, the background light contribution is much more difficult to remove, and one or more filters (SP) with substantial attenuation yet low loss (<1%) are necessary.

Instead of utilizing the intracavity circulating power, it is also possible to use the light put out by the laser and recirculate it in a secondary cavity. This cavity might consist of a standing-wave or traveling-wave *resonant* structure, or it may be built as a *non-resonant* multipass cell. Figure 4c illustrates the latter configuration. In this arrangement, two concave mirrors are positioned in a near-concentric manner to produce two regions at which the beams cross, and the photon fluxes associated with each beam add up incoherently to produce an effective circulating power that exceeds the input power by several orders of magnitude. This multipass arrangement is well known and is routinely implemented with a side detection, as depicted in Figure 4c [1]. Multipass cavities also have a long history in absorption spectroscopy, where they continue to be researched for pathlength maximization [28].

Of the three configurations depicted in Figure 4, option (b) yields the most effective utilization of the diode-pumped Pr:YLF laser for generating spontaneous Raman scattering from a gas sample. With 8 W of electrical input power, 16 W of circulating power in the forward and backward directions are available for SRS scattering. Moreover, this configuration can offer a compact and robust sensing platform.

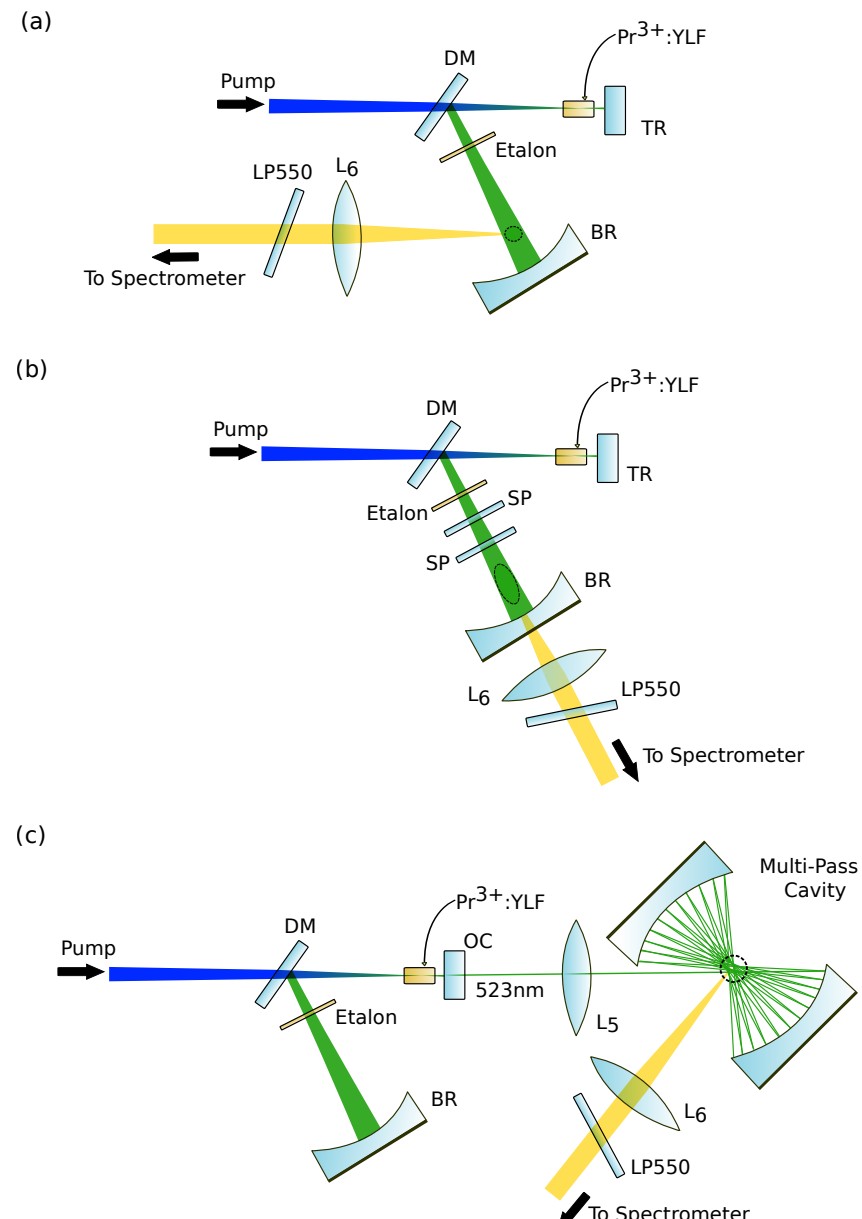

**Figure 4.** Raman enhancement configurations comparison. (**a**) Intracavity enhancement with orthogonal excitation/detection. Because no output is required, the cavity output coupler can be replaced by a total reflector (TR). (**b**) Intracavity enhancement with collinear detection. Cascaded short pass filters (SP) reject light at the Raman-shifted frequencies but transmit the laser light with as little loss as possible. (**c**) External multipass enhancement with collinear detection. In all three configurations, a collection lens ($L_6$) collimates the Raman emission, and a tilted 550 nm longpass filter (LP550) serves to reject residual light at the laser frequency. The dashed circle/ellipse is meant to indicate the volume from which the emission is collected.

### 4.2. Collinear Multipass SRS Enhancement

For simplicity, however, we opted to implement here a 4th SRS enhancement variant, illustrated in Figure 5. This variation of the Figure 4c setup uses collinear detection instead of side detection and functions as a proof-of-principle implementation; it allows the verification of high-resolution spectroscopy at concentrations comparable to those attainable with the setup of Figure 4c.

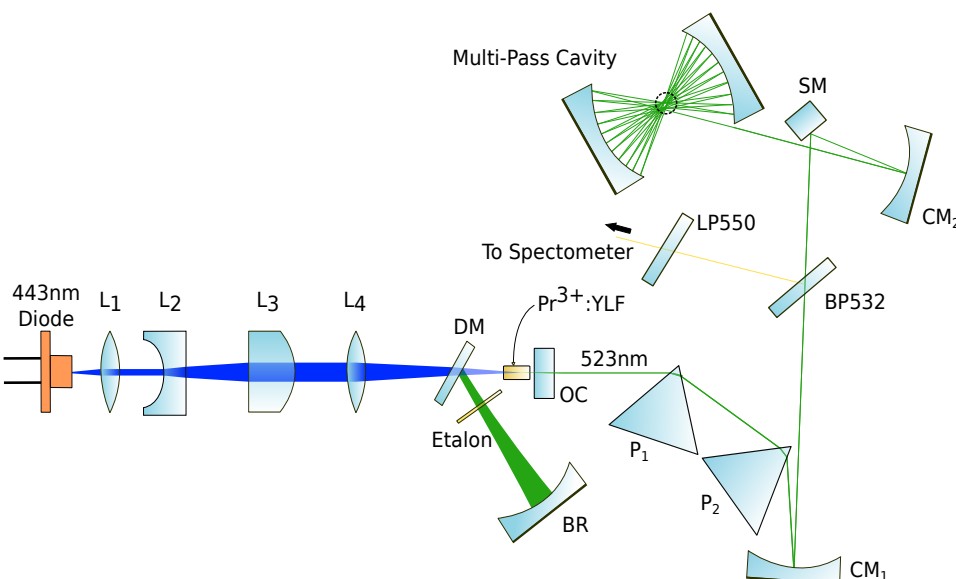

**Figure 5.** External multipass enhancement configuration with collinear detection. Dispersion prisms $P_1$ and $P_2$ remove fluorescence at the Raman-shifted frequencies. An angled 532 nm bandpass filter (BP532) transmits light at the laser frequency but reflects the collinearly returning SRS light from the multipass cavity. To focus into and collimate out of the latter, a curved mirror ($CM_2$) at near-normal incidence is employed together with a square mirror (SM). The multipass cavity consists of two opposed 25 mm diameter concave mirrors (50 mm radius of curvature) separated by 97 mm.

As seen in Figure 5, the light emerging from the Pr:YLF laser was first filtered by a pair of equilateral prisms. The prisms, which the beam entered near Brewster's angle, caused dispersion that rejected any fluorescence or other unwanted light at or near the SRS Stokes wavelengths ($\lambda \gtrsim 530$ nm). The beam was then collimated by a spherical mirror with a low angle of incidence to avoid introducing astigmatism. The beam passed through a dichroic mirror, an angled 532 nm bandpass filter (BP532) that further rejected light with $\lambda \gtrsim 530$ nm. A fold mirror directed the beam to a concave spherical mirror which focused the excitation light into the multipass cavity. The latter consisted of two identical concave high-reflectivity mirrors with a radius of curvature of 50 mm (Edmund Optics part 39–960) and reflected the pump beam about 15 times at each mirror resulting in a pathlength of several meters. The multipass mirror separation was 97 mm, and the sample collection volume was ≈10 mm$^3$. SRS collection proceeded collinearly for both forward and backward scattered light by virtue of a small off-axis adjustment of the cavity mirrors. This adjustment was such that light entering the multipass cavity eventually retraced its path. A slight angle was, however, purposefully maintained so as to avoid direct feedback to the laser cavity that would have disrupted the laser operation and changed its spectral output characteristics. As the Raman scattered light exited the multipass cavity, it was collimated by the same concave spherical mirror used for focusing the pump. The SRS light was separated dichroically from the pump, and the latter was removed by a longpass filter. Lastly, the SRS light was dispersed and imaged by a grating spectrometer with a resolution of <1 cm$^{-1}$.

### 4.3. Trace Spectroscopy of Air Samples

To examine the spectral performance of the apparatus, various gas samples were introduced into the multipass cavity. A plastic enclosure was placed over the multipass cavity to retain the samples. The enclosure had a ≈5 mm diameter wide opening for the input/output of light and a gas sample delivery via a tube-fitted port. It was thus not hermetically sealed, and all measurements were performed at atmospheric pressure and ambient temperature (22 °C), at a relative humidity of approximately 34%. Consequently, the dominant constituents of the gas sample were always nitrogen and oxygen, and all other

species were present only in trace quantities; any species introduced would be retained merely by slow diffusion through the small opening in the enclosure.

Figure 6 shows the SRS spectrum of ambient air only, obtained from the average of 10 spectra each recorded at an exposure (acquisition) time of 2 mins for a total measurement time of 20 mins. The spectral detection range was from 1200 $cm^{-1}$ to 2600 $cm^{-1}$. The dominant spectral features are the O ($\Delta J = -2$), Q ($\Delta J = 0$), and S ($\Delta J = +2$) branches of oxygen and nitrogen, as indicated. Carbon dioxide Q branches and hot bands are also visible in this range, as are the rotationally-resolved peaks associated with the bending mode of water vapor. Furthermore, peaks associated with $^{14}N^{15}N$ and $^{16}O^{18}O$ isotopologue Q-branches are clearly distinguished. As seen in the inset of Figure 6, even features associated with the oxygen nuclear spin interaction are resolved (satellite peaks indicated by arrows in the inset) [29]. The multipass cavity plays a vital role in exhibiting these detailed spectral features; in its absence, the SRS rates would be too low, as can be seen in the gray trace in Figure 6, which corresponds to a measurement in which the multipass beam was interrupted. The enhancement provided by the multipass cavity was 183-fold.

The high resolution afforded by the frequency-stabilized Pr:YLF laser lends itself particularly well to the distinction of species with nearly overlapping bands. This is the case, for example, for nitrous oxide ($N_2O$) with a Q-branch peak at 1284 $cm^{-1}$ overlapping substantially with the 1285 $cm^{-1}$ Q-branch peak of $CO_2$ (lower frequency branch of the Fermi dyad). Another strong band of $N_2O$ at 2224 $cm^{-1}$ overlaps with nitrogen O-branch peaks, but we focus here specifically on the $N_2O/CO_2$ interference. Figure 7 shows, as an overview, the SRS spectrum of ambient air (black trace) and the SRS spectrum of ambient air containing $N_2O$ at a concentration of around 0.6% (blue trace). In the inset (Figure 7a), a zoomed view of the indicated spectral region shows additional spectra in which the $N_2O$ concentration is 320 ppm and below. At such concentrations, it becomes difficult to distinguish the $CO_2$ and $N_2O$ peaks. Upon reduction with the ambient air spectrum, however, as shown in Figure 7b, it becomes clear that a shift of about 1 $cm^{-1}$ separates the two. In this manner, an $N_2O$ concentration as low as 70 ppm was measured [cyan trace in Figure 7b], and somewhat lower levels are likely detectable with increased exposure time.

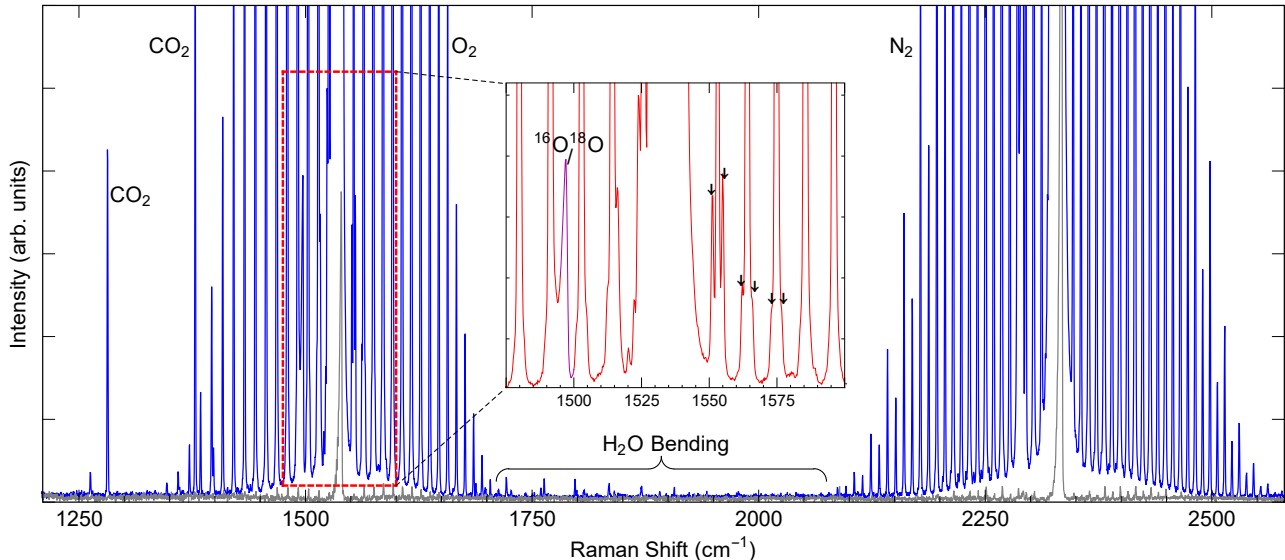

**Figure 6.** High-resolution SRS spectrum of ambient laboratory air (total measurement time of 20 mins). The inset shows a zoomed-in view of the indicated spectral region. The arrows indicate spectral features associated with the nuclear spin interaction in $O_2$. The gray trace represents a measurement in which the multipass cavity path was interrupted, resulting in only a single pump pass through the focus and backward SRS collection.

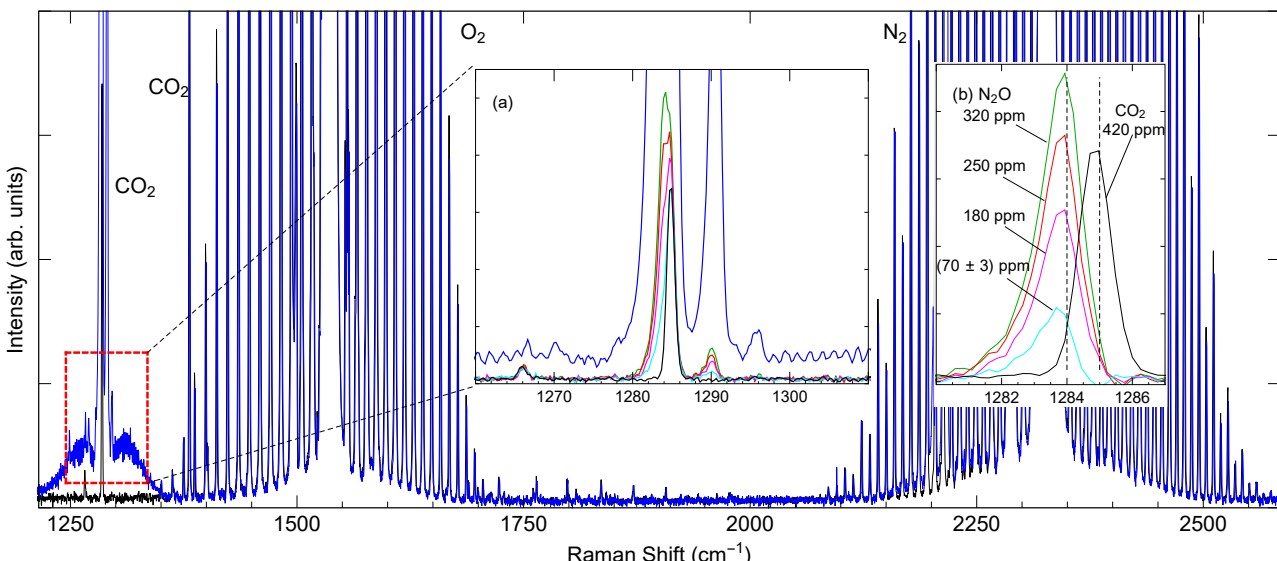

**Figure 7.** High-resolution SRS spectrum of $N_2O$ in air (blue) together with the SRS spectrum of ambient air without $N_2O$ (black). (**a**) Zoomed view with additional spectra taken at lower $N_2O$ concentrations. (**b**) Difference between the traces in part (**a**) in which $N_2O$ was present (blue, green, red, pink, and cyan) and the trace associated with ambient air without $N_2O$ (black).

A similar overlap of bands occurs for the Q-branch peak of carbon monoxide and O-branch peaks of $N_2$. A sample of car exhaust gas was introduced into the enclosure to examine the degree of overlap. The results are shown in Figure 8. As can be seen, the CO band indeed falls nearly—though not exactly—onto one of the rotationally-resolved peaks of nitrogen. It can be distinguished so long as the CO concentration is at least 20 ppm (Figure 8b pink trace). However, this would not be possible with poorer spectral resolution.

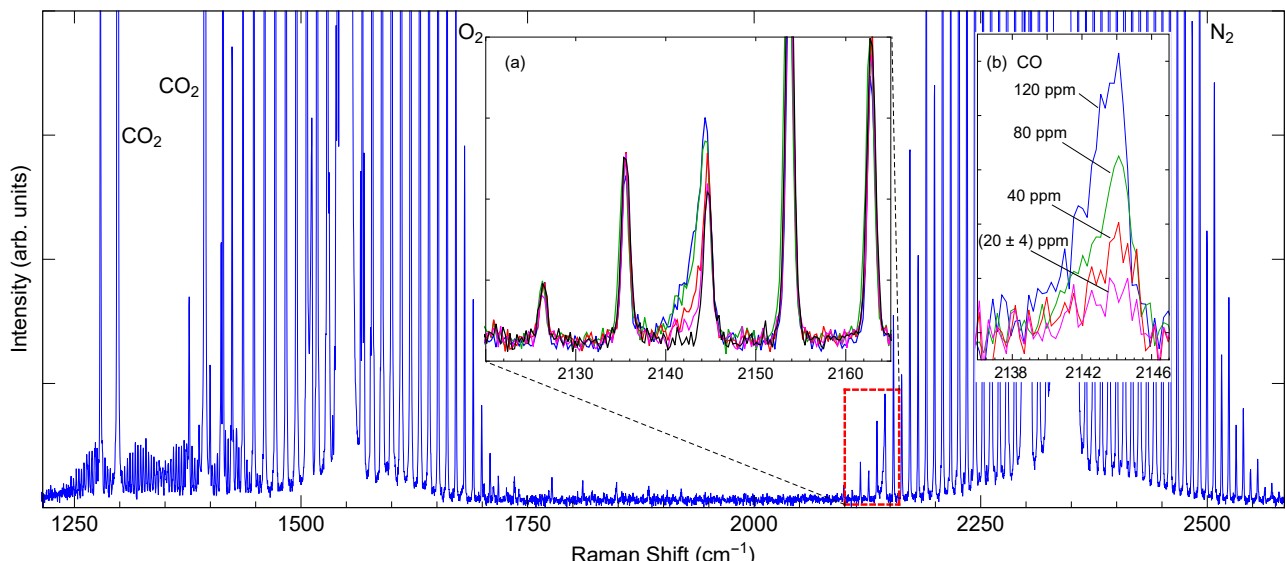

**Figure 8.** High-resolution SRS spectrum of a sample of car exhaust gas containing, in particular, CO and $CO_2$. (**a**) Magnified view, which includes spectra associated with lower concentrations of CO. (**b**) Spectra from part (**a**) from which the ambient air spectrum was subtracted.

## 5. Discussion

The capabilities demonstrated above are relevant for a number of applications. The trace detection of nitrous oxide, for example, is an essential requirement in atmospheric studies. Nitrous oxide is currently found in ambient air at an average concentration of

330 parts per billion (ppb), substantially exceeding pre-industrial levels. As a potent greenhouse gas, its atmospheric chemistry is of great interest, as is the identification and characterization of $N_2O$ sources and sinks. For this reason, accurate, sensitive, and portable sensors are critical. Currently, the tool of choice uses infrared absorption, often in the form of cavity-enhanced spectroscopy [30]. So far, these techniques are considerably more sensitive than the SRS spectroscope, with limits of detection in the ppb range and below. However, improvements in SRS enhancement methods have also brought LODs into the ppb range for some analytes. For example, methane can be detected by cavity-assisted multipass SRS at less than 20 ppb under a pressure of 0.2 MPa [18]. Similarly, hydrogen may be detected at levels of tens of ppb [31]. The scattering cross section for the main $N_2O$ band at 1284 cm$^{-1}$ is smaller than that of methane but greater than that of hydrogen. Therefore, if it were not for interference with the $CO_2$ band at 1285 cm$^{-1}$, $N_2O$ would be detectable at concentrations well below 330 ppb. The spectral discrimination demonstrated here is a step forward in overcoming this interference.

The ability to detect carbon monoxide at trace concentrations is evidently of great significance, too, for example, in pollutant monitoring, but also in medical health diagnostics such as breath analysis. The Raman scattering cross-section of CO is somewhat lower than that of, e.g., hydrogen. Nevertheless, the primary hurdle in detecting CO at sub-ppm concentrations in air is the proximity of nitrogen O-band peaks.

Improvement of spectral discrimination is possible by further reducing the laser linewidth, accompanied by a higher spectral detection resolution. Ideally, both should be near 0.1 cm$^{-1}$ to fully resolve individual rotational lines at atmospheric pressure. An additional suppression of spectral interferences can be obtained in some cases with polarization-dependent detection. In the present work, the excitation laser was highly linearly polarized, and no polarization filter was applied to the detection channel. However, the spectrometer used had a response that was highly polarization-dependent, making our measurement de facto a co-polarized SRS detection. Nonetheless, by performing additional measurements of the orthogonally-polarized Raman emission, separate isotropic and anisotropic SRS spectra can be constructed [32,33]. The isotropic CO Q-branch band would then be fully distinguishable from the anisotropic nitrogen O-branch lines. Polarization-resolved detection would also benefit the measurement of $N_2O$ in the presence of $CO_2$ as in Figure 7, by removing overlap with $CO_2$ O-branch peaks near 1284 cm$^{-1}$, which can be seen in the data of Figure 8 where the $CO_2$ concentration is high.

Ultimately the ability to detect at lower concentrations, i.e., in the ppb range, will be limited by SRS rates. These can be augmented considerably compared to the rates obtained in the present experiments. Pressurization of the gas sample is the most straightforward route to achieve a rate increase [34]. Beyond pressurization, optimal utilization of laser power is the key factor in enabling scalable improvements in sensitivity. The linearity of Raman scattering and the fact that the laser power loss in gas SRS is negligible compared to, e.g., optical losses in mirrors, implies that significant gains can still be made in SRS sensitivity.

## 6. Conclusions

In conclusion, we explored the usage of a Pr:YLF laser with narrowed spectral bandwidth for high-resolution SRS spectroscopy. Such a laser is advantageous for Raman scattering in gases as it does not generate laser light by way of a second-harmonic conversion, as conventional high-power visible narrow-band light sources do. Besides improved laser efficiency, the high circulating intracavity power can be directly utilized for SRS enhancement. The focus here was on a relatively low-power regime in which only a single, passively-cooled InGaN laser diode served as the pump source. Even so, our measurements show that remarkable spectral discrimination can be obtained at trace concentrations and atmospheric pressure, by simply using a multipass arrangement. Given that the power scaling of Pr:YLF lasers has already been demonstrated, it is reasonable to anticipate that considerably lower limits of detection than those reported here will be realized. Further-

more, it is conceivable that multipass enhancement can be combined with circulating power enhancement if technical challenges associated with optical alignment and optomechanical stability can be overcome.

**Author Contributions:** Conceptualization, A.M.; Methodology, A.M.; Software, A.M.; Validation, A.M. and C.M.A.; Formal Analysis, C.M.A.; Investigation, C.M.A.; Resources, A.M. and C.M.A.; Data Curation, C.M.A.; Writing—Original Draft Preparation, A.M.; Writing—Review and Editing, A.M. and C.M.A.; Visualization, C.M.A.; Supervision, A.M.; Project Administration, A.M.; Funding Acquisition, A.M. All authors have read and agreed to the published version of the manuscript.

**Funding:** National Science Foundation (NSF grant No. 2116275).

**Institutional Review Board Statement:** Not applicable.

**Informed Consent Statement:** Not applicable.

**Data Availability Statement:** The data presented in this study are available on request from the corresponding author.

**Conflicts of Interest:** The authors declare no conflict of interest.

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
