# Peer review of "Narrow-Linewidth Pr:YLF Laser for High-Resolution Raman Trace Gas Spectroscopy"

_2813-446X, doi:10.3390/spectroscj1020008_

Round 1

Reviewer 1 Report

1-      The current work's novelty, what makes it special, and how it varies from prior studies should all be explained by the authors.

2-      Please describe the multipass cavity parameters in the text, including volume and achievable optical path length.

3-      Please use the dimensions of the multipass cavity and the absorption cross-section to determine the sensitivity limit for the trace gasses.

4-      Can a graph that compares the SRS spectrum while employing a multipass cavity to a single-pass cavity be added to the manuscript?

5-      Nothing is mentioned in the text about the uncertainty in the measurements, so please discuss this issue in the text.

6-      In addition to the References list, other current research is proposed to be reviewed and included if it is beneficial:

Ø    “Multipass cell based on confocal mirrors for sensitive broadband laser spectroscopy in the near infrared”. Appl Opt. 2013 Oct 10;52(29):7145-51. doi: 10.1364/AO.52.007145.

Overall, the manuscript appears to be well-written

Author Response

1-      The current work's novelty, what makes it special, and how it varies from prior studies should all be explained by the authors.

Response: The main novelty in the approach we describe is the utilization of a solid-state laser source for Raman gas spectroscopy that does not generate laser light by way of second-harmonic generation as conventional high-power visible narrow-band light sources do. As we explain in the introduction, this is consequential because it makes it possible to utilize the high intracavity power for Raman enhancement. It also provides a portable solution because of greater efficiency of the laser. What we show here for the first time is that the Pr:YLF laser can have spectral characteristics suitable for high-resolution Raman gas spectroscopy. We have added a paragraph in the conclusion to reiterate this point further.

2-      Please describe the multi-pass cavity parameters in the text, including volume and achievable optical path length.

Response: We have added further specifications associated with the multipass cavity. We note that in contrast to absorption spectroscopy, the path length is not a critical parameter for Raman spectroscopy. What matters most is how many times the beam re-traverses the same focal volume. In that regard, the volume being probed (here about 10 cubic mm) is much smaller than the physical volume of the cell defined by the mirror diameter and separation (close to 50000 cubic mm).

3-      Please use the dimensions of the multi-pass cavity and the absorption cross-section to determine the sensitivity limit for the trace gasses.

Response: In contrast to absorption spectroscopy, the absorption of the gas sample does not affect the sensitivity of the measurement insofar as the pump beam does not get measurably depleted by the Raman scattering process.

4-      Can a graph that compares the SRS spectrum while employing a multi-pass cavity to a single-pass cavity be added to the manuscript?

Response: Yes, we welcome the suggestion and have added such a graph in Fig. 6. The multi-pass cavity provides an enhancement of approximately 183 fold.

5-      Nothing is mentioned in the text about the uncertainty in the measurements, so please discuss this issue in the text.

Response: We have added a comment on measurement uncertainties and given representative figures.

6-      In addition to the References list, other current research is proposed to be reviewed and included if it is beneficial:

Ø    “Multipass cell based on confocal mirrors for sensitive broadband laser spectroscopy in the near infrared”. Appl Opt. 2013 Oct 10;52(29):7145-51. doi: 10.1364/AO.52.007145.

Response: We have included the reference in the text and noted that multi-pass cells have been and are being thoroughly researched for gas absorption spectroscopy.

Reviewer 2 Report

This paper is important to the SRS gas spectroscopy community and can be considered for publication after the authors make changes in the paper with regards to the following issues:

 1. In line 100, the authors need to illustrate the laser threshold.

 2. In Table 1, the “Pump beam exp.” should be “Pump beam expander”.

 3. In Figure 3, the “microns” should be “ ”.

 4. In line 142, “the beam’s profile is somewhat elliptical…”, the laser beam could be improved     with fiber-coupled 443 nm pump diode, the authors need to present the explanation.

 5. In Figure 4, the dashed circles indicate the localized volume of gas, the authors need to add the statement in caption.

 6. Compare to multipass enhancement configuration, the intracavity enhancement configuration is the most effective utilization of the diode-pumped Pr:YLF laser for SRS gas detection. Why do the authors opted to the collinear multipass enhancement configuration?

7.  In the caption of Figure 5, the “An angled 532 nm bandpass-filter…” should be “An angled 532 nm bandpass-filter (BP532)…”.

8.  In Figure 7 (b), the black trace for 420 ppm CO2 is different from the one in Figure 7 (a), the authors need to explain the appearance.

9. In the caption of Figure 7, the “N2O was present (blue, green, …” should be “N2O was present (green, …”. 

please check some minor grammar and textual expression

Author Response

This paper is important to the SRS gas spectroscopy community and can be considered for publication after the authors make changes in the paper with regards to the following issues:

      1. In line 100, the authors need to illustrate the laser threshold.

Response: We now provide the threshold (0.9 W) in the text.

       2. In Table 1, the “Pump beam exp.” should be “Pump beam expander”.

Response: We have removed the abbreviation.

       3. In Figure 3, the “microns” should be “ ”.

Response: We assume the referee is referring to the “um” unit (in the viewable document the quotes are empty) and we have made the change.

      4. In line 142, “the beam’s profile is somewhat elliptical…”, the laser beam could be improved     with fiber-coupled 443 nm pump diode, the authors need to present the explanation.

Response: We agree with the referee. The output beam is slightly elliptical most likely because of the astigmatism of the pump beam. The pump beam astigmatism could be removed, for example by an intermediary fiber coupling, or the output beam asymmetry could be compensated by, e.g., a prism pair, if needed. We have added text to qualify our original statement.

     5. In Figure 4, the dashed circles indicate the localized volume of gas, the authors need to add the statement in the caption.

Response: We thank the reviewer for this suggestion and regret the omission. We have now added an appropriate comment in the caption.

     6. Compare to the multi-pass enhancement configuration, the intracavity enhancement configuration is the most effective utilization of the diode-pumped Pr:YLF laser for SRS gas detection. Why do the authors opted to the collinear multi-pass enhancement configuration?

Response: The intracavity enhancement is the most effective enhancement approach in the sense of power scaling and compactness. There is no requirement on extra space and optics that the multi-pass cavity approach needs. The intracavity power can be raised (in principle to hundreds of watts or more, limited principally by intracavity loss) as long as the intracavity beam diameter is not too small to avoid thermo-optic effects. However, this approach requires, among other specialized components, expensive filters that were not available to us. The multi-pass approach allows us to create a large enhancement of about 180 (see added data to Fig. 6 requested by referee 1) while still demonstrating the functionality provided by the Pr:YLF laser for gas spectroscopy.

     7. In the caption of Figure 5, the “An angled 532 nm bandpass-filter…” should be “An angled 532 nm bandpass-filter (BP532)…”.

Response: We thank the reviewer for pointing this out and we have made the correction.

    8. In Figure 7 (b), the black trace for 420 ppm CO2is different from the one in Figure 7 (a), the authors need to explain the appearance.

Response: The black traces represent the ambient air measurement. The data associated with the black trace is the same in Fig. 7(a) and Fig. 7(b) but the scales used in the graphs are different (the frequency range is smaller in (b)).

       9. In the caption of Figure 7, the “N2O was present (blue, green, …” should be “N2O was present (green, …”. 

Response: There may be a confusion here between the blue and black traces. As indicated in the figure caption, the black trace represents ambient air (no added N2O) while all the other traces represent ambient air with a small amount of added N2O. For the blue trace, this amount is about 0.6% while for all the other traces the N2O concentration was in the parts-per-million range as indicated in Fig. 7(b).

Reviewer 3 Report

The paper is devoted to the development of a laser source for Raman spectroscopy. In the paper, a laser source with a narrow spectrum is developed, and its application in various optical schemes for the analysis of various gases is demonstrated. The presented results have scientific novelty and practical value.

All the results are clearly presented and are beyond doubt. The paper can be published in its current view.

Author Response

The paper is devoted to the development of a laser source for Raman spectroscopy. In the paper, a laser source with a narrow spectrum is developed, and its application in various optical schemes for the analysis of various gases is demonstrated. The presented results have scientific novelty and practical value.

All the results are clearly presented and are beyond doubt. The paper can be published in its current view.

Response: We thank the reviewer for the positive assessment.

Round 2

Reviewer 1 Report

The authors have made reasonable changes to the manuscript in response to my previous suggestions and concerns. In my opinion, the manuscript now has all information and is ready for publication as a regular article in the Journal " Spectroscopy Journal ".

Overall the manuscript appears to be clearly and carefully written.